# Bone Mass and Osteoblast Activity Are Sex-Dependent in Mice Lacking the Estrogen Receptor α in Chondrocytes and Osteoblast Progenitor Cells

**DOI:** 10.3390/ijms23052902

**Published:** 2022-03-07

**Authors:** Lena Steppe, Jasmin Bülow, Jan Tuckermann, Anita Ignatius, Melanie Haffner-Luntzer

**Affiliations:** 1Institute of Orthopedic Research and Biomechanics, University Medical Center Ulm, 89081 Ulm, Germany; lena.steppe@uni-ulm.de (L.S.); jasmin.buelow@uni-ulm.de (J.B.); anita.ignatius@uni-ulm.de (A.I.); 2Institute of Comparative Molecular Endocrinology (CME), Ulm University, 89081 Ulm, Germany; jan.tuckermann@uni-ulm.de

**Keywords:** bone, estrogen, receptor knockout, genetic animal model, sex steroids, osteoblasts, chondrocytes, biomechanics

## Abstract

While estrogen receptor alpha (ERα) is known to be important for bone development and homeostasis, its exact function during osteoblast differentiation remains unclear. Conditional deletion of ERα during specific stages of osteoblast differentiation revealed different bone phenotypes, which were also shown to be sex-dependent. Since hypertrophic chondrocytes can transdifferentiate into osteoblasts and substantially contribute to long-bone development, we aimed to investigate the effects of ERα deletion in both osteoblast and chondrocytes on bone development and structure. Therefore, we generated mice in which the ERα gene was inactivated via a *Runx2*-driven cyclic recombinase (ERα^fl/fl;^ ^Runx2Cre^). We analyzed the bones of 3-month-old ERα^fl/fl;^ ^Runx2Cre^ mice by biomechanical testing, micro-computed tomography, and cellular parameters by histology. Male ERα^fl/fl;^ ^Runx2Cre^ mice displayed a significantly increased cortical bone mass and flexural rigidity of the femurs compared to age-matched controls with no active Cre-transgene (ERα^fl/fl^). By contrast, female ERα^fl/fl;^ ^Runx2Cre^ mice exhibited significant trabecular bone loss, whereas in cortical bone periosteal and endosteal diameters were reduced. Our results indicate that the ERα in osteoblast progenitors and hypertrophic chondrocytes differentially contributes to bone mass regulation in male and female mice and improves our understanding of ERα signaling in bone cells in vivo.

## 1. Introduction

Sex steroids are essential for skeletal development and regulate bone mass in both males and females [1]. During puberty, estrogen exerts suppressive effects on bone growth, whereas testosterone promotes periosteal bone formation resulting in greater bone size and a stronger skeleton in males [2]. Highlighting the important functions of both hormones on skeletal maintenance, declining levels of estrogen during menopause, and a reduction of both estrogen and androgens with increasing age in men are associated with bone loss and the development of osteoporosis [3,4,5]. The effects of estrogen on bone are mainly mediated via the two estrogen receptors (ERs), ERα and ERβ, which can both be also activated by androgens after aromatization into estrogens [6,7]. In bone, ERα and ERβ are expressed in chondrocytes, osteoblasts, osteocytes, and osteoclasts [8,9]. It is known that ERα is expressed at higher levels in cortical bone, whereas ERβ is strongly expressed in cancellous bone [10]. Because ERβ modulates the transcriptional activity of ERα and antagonizes many of ERα-driven effects [11,12], a “ying yang” relationship between both receptors is described [13]. ERα has become a major focus of research, since a human case report demonstrated that a point mutation in its gene resulted in osteoporosis and unfused growth plates in a 28-year-old man [14]. Furthermore, case reports from men that suffer from an aromatase deficiency together with preclinical studies demonstrated the importance of estrogens for male skeletal homeostasis [15,16,17,18]. In bone, ERα is mainly mediating the actions of estrogen and is therefore considered as more important for skeletal homeostasis [19,20]. Regarding sex-specific differences of ERα expression, we showed that during bone healing, ERα expression is lower in the fracture callus of male mice, specifically in hypertrophic chondrocytes, shown by immunohistochemical stainings and gene expression analysis of the fractured femurs [21]. Furthermore, in the literature, it is reported that cultured human osteoblasts from men and women respond differentially to estrogen or ERα/ERβ-specific agonists in a sex-specific manner [22,23]. These findings indicate that the expression levels of ERα/ERβ as well as their response to its ligand estrogen might be sex-specific. For this reason, we focused on the sex-specific role of ERα on bone.

The crucial role of ERs in skeletal health and growth was further confirmed by preclinical studies using global ERα- (ERα-KO) or ERβ-knockout (ERβ-KO) mice. In females and males, ERα-KO resulted in shorter long bones, whereas ERβ-KO showed no effects on longitudinal bone growth [16,24,25,26]. Additionally, ERα-KO mice displayed a decrease in cortical bone mineral density (BMD) and an increased trabecular bone volume in both sexes [27]. However, in a mouse line with a global deletion of the ERα, unspecific effects or compensatory mechanisms by the increased serum levels of testosterone [27] or the low expression levels of a chimeric truncated ERα isoform due to alternative splicing [28] cannot be ruled out. Therefore, cell-specific knockout models were generated using the Cre/loxP technology to elucidate the role of ERα on different stages of the osteoblast life cycle (Figure 1, Table 1). Under the control of different promotor genes particularly expressed by the respective cell type, the expression of a Cre-recombinase leads to a deletion of the ERα in this cell type.

Regarding the differentiation timeline of osteoblasts (Figure 1), first, *Prx1* is expressed by skeletal stem cells followed by Runx2, which is upregulated in osteoblastic precursors together with *Col1a1* [36]. When these precursors further differentiate, osterix (*Osx1*) is upregulated in pre-osteoblasts which then become mature osteoblasts, expressing high levels of osteocalcin (*OC*). *Dmp1* is mainly expressed by osteocytes. ERα deletion in skeletal stem cells (*Prx1*-Cre) resulted in a reduction of cortical bone mass and BMD in both sexes, whereas the trabecular bone mass was unchanged [29]. When ERα was deleted in pre-osteoblasts (*Osx1*-Cre), female mice displayed a reduced cortical thickness. A deletion of the ERα during the matrix maturation phase in osteoblasts via the *Col1a1*-Cre model [29] had no effect on the bone phenotype, neither in male nor in female mice. However, ERα deletion in mature osteoblasts and osteocytes (*OC*-Cre) resulted in a reduction of cortical and trabecular bone mass in females but not in males [32,33,34].

Taken together, these studies have established that bone development is regulated by ERα in osteoblastic cells. Thereby, ERα-mediated effects are dependent on the osteoblast differentiation stage and on the sex [32,33,34]. However, these studies did not take into account that hypertrophic chondrocytes can undergo transdifferentiation into osteoblasts [37,38], which essentially contributes to bone formation in long bones [39,40]. Therefore, here we investigated the role of ERα in both osteoblast lineage-specific cells and hypertrophic chondrocytes by using the *Runx2*-Cre mouse model. The transcription factor *Runx2* is essential for driving osteoblastogenesis and is expressed in hypertrophic chondrocytes, eventually transdifferentiating into osteoblasts (Figure 1) [41,42,43]. We crossbred *Runx2*-Cre with ERα floxed mice (ERα^fl/fl^) and analyzed the bone phenotype in both male and female mice to provide further insights into the regulatory functions of ERα in bone development.

We analyzed bone strength of male and female 3-month-old ERα^fl/fl; Runx2Cre^ mice by biomechanical testing, bone mass and structure by μCT, and cellular parameters by histology. In male ERα^fl/fl; Runx2Cre^ mice, bone strength and cortical thickness were significantly increased compared to age-matched floxed control littermates with no active Cre-transgene (ERα^fl/fl^). In addition, the periosteal diameter and the moment of inertia were found to be significantly increased, whereas the ERα knockout had no effect on trabecular bone. By contrast, female ERα^fl/fl; Runx2Cre^ mice exhibited significant trabecular bone loss, whereas the cortical bone was mainly unaffected. Our results indicate a sex-specific role of the ERα in osteoblast progenitors and hypertrophic chondrocytes on bone mass in male and female mice.

## 2. Results

### 2.1. Influence of ERα Deletion in Osteoblast Progenitors on Bone in Male Mice

The femur lengths of male ERα^fl/fl; Runx2Cre^ mice were not different compared to ERα^fl/fl^ control animals (Figure 2A). To evaluate the flexural rigidity of the femurs and the bone structure of ERα^fl/fl; Runx2Cre^ mice, biomechanical testing and μCT analyses were performed. The ERα^fl/fl; Runx2Cre^ males displayed a significant increased flexural rigidity of the femurs compared to their control littermates (Figure 2B), resulting from a significantly higher moment of inertia and from a significantly increased cortical TMD and thickness (Figure 2C–E). The periosteal bone diameter was significantly increased in ERα^fl/fl; Runx2Cre mice^, whereas the endosteal diameter was unchanged between both genotypes (Figure 2F,G). In the trabecular compartment, all analyzed bone parameters including BV/TV, trabecular number, thickness, and separation, as well as trabecular BMD, were unaltered (Figure 2H–L).

Histomorphometric analyses of the metaphyseal region demonstrated that trabecular osteoblast number and surface were not affected by the ERα knockout (Figure 3). Similarly, trabecular osteoclast number and surface, as well as cortical osteocyte number, were unaltered (Figure 3). At the growth plate, osteoclast number as well as growth plate thickness were unchanged between both genotypes (Appendix A). Taken together, these results suggest that the deletion of ERα provoked an anabolic effect on the cortical bone of male mice.

### 2.2. Influence of ER α Deletion in Osteoblast Progenitors on Bone in Female Mice

In contrast to male ERα^fl/fl; Runx2Cre^ mice, the femur lengths, flexural rigidity and the moment of inertia did not significantly differ in female ERα^fl/fl; Runx2Cre^ mice compared to control animals (Figure 4A–C). Periosteal and endosteal diameters of the femurs were significantly reduced in female ERα^fl/fl; Runx2Cre^ mice, whereas the cortical TMD and thickness were unaltered (Figure 4D–G).

In trabecular bone, the lack of the ERα in female mice led to severely reduced BV/TV, BMD, trabecular number, and thickness (Figure 4H–J,L,N and Appendix A for epiphyseal bone). Consistently, trabecular separation was significantly increased in ERα^fl/fl; Runx2Cre^ animals in comparison to their ERα^fl/fl^ littermates (Figure 4K). Furthermore, a significant reduction of the osteoblast number and surface in the trabecular bone was observed in female ERα^fl/fl; Runx2Cre^ mice, whereas osteoclast number and surface were unaltered (Figure 5A–H). In addition, we found a significantly reduced osteocyte number per bone area in the cortical bone of ERα^fl/fl; Runx2Cre^ animals (Figure 5I). Furthermore, at the growth plate, osteoclast number and growth plate thickness were not affected by the knockout (Appendix A). Osteogenic differentiation experiments of primary osteoblasts cultures isolated from both genotypes revealed a significantly diminished osteogenic differentiation ability of cells derived from ERα^fl/fl; Runx2Cre^ animals shown by a significantly lower gene expression of alkaline phosphatase (*AP*), osteocalcin (*OC*), bone sialoprotein (*BSP*), and runt-related transcription factor-2 (*Runx2)* in vitro and shown by weaker AP stainings (Figure 5J–Q). Collectively, these findings indicate that female ERα^fl/fl; Runx2Cre^ animals exhibited a reduced trabecular bone mass and altered cortical bone architecture based on a reduced osteoblast differentiation.

## 3. Discussion

In the present study, we aimed to examine whether ERα signaling in osteoblast lineage-specific cells and hypertrophic chondrocytes are required for bone formation and development. To this end, mice with a deletion of the ERα in both cell types were generated by breeding *Runx2*-Cre and ERα floxed mice. The *Runx2*-Cre model allows to study the role of ERα not only in osteoblast progenitor cells directly derived from skeletal stem cells, but also in hypertrophic chondrocytes eventually transdifferentiating into osteoblasts during long bone development.

In male ERα^fl/fl; Runx2Cre^ mice, flexural rigidity and cortical thickness of the femurs were significantly increased, while the trabecular bone mass was not affected.

In contrast to the cortical bone phenotype found by our study, others observed a decrease in cortical bone mass which could be attributed to the used *Prx1*-Cre model, leading to a deletion of the ERα very early in skeletal stem cells [29]. No effects on either bone mass or composition were observed when using the *Col1a1*-Cre model to delete the ERα [29]. However, similar to our results, cortical bone mass and biomechanical strength was increased in male mice when ERα was deleted in mature osteoblasts and osteocytes (*OC*-Cre) [33]. However, these mice also displayed an increased femoral length as well as a significantly higher cancellous BV/TV and greater trabecular thickness at the tibial metaphysis [33]. Most likely, these differences are due to the different Cre model that was used. In other studies that also used the *OC*-Cre model, no cortical or trabecular changes were found in 16-week-old male mice [34] or in 26-week old mice [31], whereas in 24-week-old male mice, tibial BV/TV and TbN were lower [34]. However, the genetic background of those mice was not reported [34] while in the study of Melville and in our study, C57BL/6 mice were used. The genetic background can strongly influence bone structure and mass [44], which might explain these inconsistent results. A study by Windahl and colleagues analyzed the bone phenotype of male mice with an ERα deletion in osteocytes, showing that the trabecular bone volume was significantly reduced while cortical bone was unaffected [35]. The authors propose that the ERα in osteocytes is necessary to facilitate estrogen signaling that may result in an enhanced osteoblast activity which might explain the reduced trabecular bone phenotype due to a reduced osteoblast activity [35].

To draw conclusions about the role of the ERα on hypertrophic chondrocytes, we compare our results to the data generated under the *Col1a1*-Cre promotor, because it is known that *Col1a1* and *Runx2* are simultaneously upregulated in osteoblast precursors. *Runx2* is known to induce *Col1a1* gene expression in osteoblasts and is further expressed by transdifferentiating hypertrophic chondrocytes [41]. In contrast to the male *Col1a1*-Cre mice, which did not display any bone phenotype, we observe in our *Runx2*-Cre animals a significantly increased cortical thickness and a higher flexural rigidity. These differences suggest that in *Runx2*-Cre mice, the additional deletion of the ERα in hypertrophic chondrocytes might have a crucial impact on the bone phenotype, indicating that the ERα under physiological conditions might have a regulatory role on the transdifferentiation of hypertrophic chondrocytes into osteoblasts and that this process might contribute to cortical bone accrual in male mice. Another reason for the increased cortical thickness might be the involvement of ERα in the estrogen-dependent suppression of periosteal bone mass accrual during bone development [1,5,33,45], which in addition might explain the higher cortical bone mass in male ERα^fl/fl; Runx2Cre^ mice. Furthermore, testosterone might have a stronger positive impact on periosteal bone formation in male mice lacking ERα and possibly result in an increase in cortical thickness via androgen receptors [33]. These findings indicate that ERα mediates its effects on limiting periosteal expansion primarily during the stage of *Runx2*-positive osteoblast precursor cells, as well as during the *OC*-positive mature osteoblasts. In trabecular bone, the effect of ERα appears less prominent, because most of the studies do not report an altered trabecular bone microarchitecture in male mice.

ERα deletion in osteoblast progenitor cells and hypertrophic chondrocytes of female mice resulted in significant trabecular bone loss, whereas the bending stiffness of the femurs and the cortical thickness were unaltered.

When ERα deletion occurred at the stage of skeletal stem cells (*Prx1*-Cre) prior to *Runx2* expression, cortical bone mass and BMD were reduced in female mice, whereas the trabecular bone mass was unchanged [29]. Others demonstrated a significantly reduced trabecular bone mass in the tibiae and spine of female *Runx2*-Cre knockout mice aged 16 weeks, confirming our results [30]. By contrast, in the *Col1a1*-Cre model, no effect on cortical or trabecular bone was observed [29]. When ERα was deleted at the stage of pre-osteoblasts (*Osx1*-Cre), female mice displayed a reduced cortical thickness and unchanged trabecular bone parameters. Other studies that have used the *OC*-Cre model to delete the ERα in mature osteoblasts and osteocytes demonstrated that the trabecular bone mass was significantly reduced in female ERα-KO mice together with an altered cancellous architecture in the proximal tibia, distal femur, and L5 vertebral body [31,32,33,34]. Additionally, they observed a significant decrease in the cortical area and cortical thickness. Interestingly, the ERα on osteocytes was not required for maintaining both trabecular and cortical bone mass in female mice [35].

In comparison to the *Col1a1*-Cre model, which showed no bone phenotype, our data suggest that the decreased trabecular bone mass of the female ERα^fl/fl; Runx2Cre^ mice results from the reduced number of trabecular osteoblasts that were also found to be significantly reduced in their surface, which is a surrogate for their activity. We hypothesize that the presence of the ERα on hypertrophic chondrocytes might be crucial for the transdifferentiation into osteoblasts that might participate in the formation of trabecular bone. However, future studies are needed to characterize the role of ERα in transdifferentiation of hypertrophic chondrocytes in more detail. Furthermore, we found that primary osteoblasts isolated from female ERα^fl/fl; Runx2Cre^ displayed a significantly reduced differentiation ability. This finding is consistent with one study showing that ERα-deficient osteoblast progenitors from the periosteum have a decreased differentiation potential [29]. In cortical bone of the female ERα^fl/fl; Runx2Cre^ mice, periosteal and endosteal diameters as well as osteocyte numbers were significantly reduced. These findings confirm that the ERα in osteoblast precursors is involved in endosteal bone resorption [46] and explain the reduced endosteal diameter. Regarding the reduced periosteal diameter, it is known that the ERα is important for periosteal bone apposition in female mice [29], resulting in a lower periosteal diameter when ERα is deleted. However, the molecular pathways behind these findings need to be addressed in further studies.

Taken together, the functions of ERα appear to be highly dependent on the maturation status of osteoblasts, while it starts to exert its positive effects on the trabecular bone of female mice at the stage of *Runx2*-positive osteoblast progenitor cells and is additionally required in mature osteoblasts, but not osteocytes. In cortical bone, ERα might be involved at very early stages of osteoblast maturation (*Prx1*-Cre) as well as in mature osteoblasts (*OC*-Cre) in female mice.

When the ERα was specifically deleted in chondrocytes by using a *Col2a1*-Cre model, no effects on skeletal growth or bone mineral density was reported [47] suggesting that the cartilage-specific ERα might not influence bone metabolism and structure. In our *Runx2*-Cre model, transdifferentiating chondrocytes and osteoblastic cells are affected by the deletion and we observed a bone phenotype in both males and females, suggesting a sex-specific role of ERα in hypertrophic chondrocytes.

One limitation of our study is that we only analyzed the bone phenotype at the age of 12 weeks, whereas it would be of interest to also include also younger and older mice to investigate bone growth and development in more detail. Furthermore, to study the contribution of estrogen-dependent or -independent signaling to the observed bone phenotype in female ERα^fl/fl; Runx2Cre^ mice, it would be of interest to investigate the bone response to ovariectomy in these mice.

In conclusion, we demonstrated that the ERα in osteoblast progenitors and hypertrophic chondrocytes is crucial for mediating anabolic effects on trabecular bone in female mice, whereas in males it has different effects by limiting the estrogen-dependent periosteal apposition in cortical bone. This suggests a sex-specific role of the ERα that is either promoting or repressing bone apposition at different sites of the long bones. The ERα in hypertrophic chondrocytes might be critically involved in the transdifferentiation towards osteoblasts, thereby regulating bone mass. These findings might be relevant also for human bone development, because case reports have already demonstrated the importance of estrogens and ERα signaling for human skeletal homeostasis [15,16,17,18].

## 4. Materials and Methods

### 4.1. Animal Care and Animal Models

All experiments were performed according the ARRIVE guidelines and were approved by the local ethical committee (o.135-9, Regierungspräsidium Tübingen, Germany). Osteoblast progenitor cell-specific ERα-KO mice (Tg(Runx2-cre)1Jtuc x Esr1tm1.2Mma) were generated by crossing Runx2-Cre with ERα^fl/fl^ mice on a C57BL/6 background. ERα^fl/fl; Runx2Cre^ mice were shown to lack the ERα in cells of the osteogenic lineage and in hypertrophic chondrocytes [48]. Age-matched, floxed littermates (ERα^fl/fl^) with no active Cre-transgene were used as control. All animals were housed in groups of up to five mice per cage with a 14-h light, 10-h dark rhythm and received water ad libitum as well as a standard mouse feed (ssniff R/M-H, V1535-300; Ssniff, Soest, Germany). At the age of 12 weeks, mice were sacrificed using an isoflurane overdose. Mouse genotyping was conducted by lysed tail PCR using the primers: 5′-CCA GGA AGA CTG CCA GAA GG-3′, 5′-TGG CTT GCA GGT ACA GGA G-3′ and 5′-GGA GCT GCC GAG TCA ATA AC-3′ to detect the Cre transgene, whereas the ERα loxP sites were detected using the primers: 5′-TAG GCT TTG TCT CGC TTT CC-3′, 5′-CCC TGG CAA GAT AAG ACA GC-3′ and 5′-AGG AGA ATG AGG TGG CAC AG-3′.

### 4.2. Biomechanical Testing

To evaluate flexural rigidity of the femora, a non-destructive three-point bending test was performed on freshly prepared samples as described previously [49]. Briefly, an axial load with a maximum of 4 N was applied on the cranial side at the diaphyseal region of the bone using a material-testing machine (1454, Zwick GmbH Co KG, Ulm, Germany). The flexural rigidity was calculated from the slope of the load-deflection curve.

### 4.3. μCT Analysis

Femora were fixed in 4% phosphate-buffered formaldehyde solution and scanned using a μCT scanning device (Skyscan 1172 v1.5; Skyscan, Kontich, Belgium) at a resolution of 8 μm using a peak voltage of 50 kV and 200 μA. Analyses and calibration steps were performed according to the guidelines of the American Society for Bone and Mineral Research [50] and volumes of interest were chosen as previously described [51]. Briefly, cortical bone was evaluated in a volume of interest (VOI) of 168 μm length within the mid-diaphyseal region. The trabecular bone was assessed 200 μm proximal of the metaphyseal growth plate over a length of 280 μm in the distal femur, excluding the cortex. For assessing the epiphyseal femur, the trabecular bone was evaluated distal from the growth plate by a defined cylindrical VOI.

Within each scan, two phantoms with a defined density of hydroxyapatite (250 and 750 mg hydroxyapatite/cm^3^) were included to determine the bone mineral density. To distinguish between mineralized and nonmineralized tissue, thresholds for cortical bone (641.9 mg hydroxyapatite/cm^3^) and trabecular bone (394.8 mg hydroxyapatite/cm^3^) were used [52]. Analyses were performed by means of Skyscan software (NRecon v1.7.1.0, DataViewer v1.5.1.2, and CTAn v1.17.2.2).

### 4.4. Histomorphometry

Femora were fixed in 4% phosphate-buffered formaldehyde solution, decalcified using 20% EDTA (pH 7.2 to 7.4) for at least 14 days, and embedded in paraffin. The number and surface of osteoclasts was evaluated using tartrate-resistant alkaline phosphatase (TRAP) staining, whereas osteoblast number and surface were analyzed using Toluidine blue-stained sections. Osteoclasts were defined as TRAP-positive cells with two or more nuclei, directly located on the bone surface with a visible resorption lacunae between the bone surface and the cell. Osteoblasts were identified as cubic-shaped and Toluidine blue-positive cells with visible cytoplasm, directly located on the bone surface. Both osteoclasts and osteoblast parameters were evaluated at the femur metaphysis and normalized to bone surface using the Osteomeasure system (OsteoMeasure system v4.1.0.0; OsteoMetrics, Inc., Decatur, GA, USA). Growth plate thickness was analyzed in Safranin O-stained sections using the Osteomeasure system. Osteoclast activity at the growth plate was assessed within a defined region of 480 μm × 350 μm and osteoclasts were identified as multinucleated cells either laying directly on bone or cartilage tissue. Images were obtained using a Leica microscope (DMI 6000B). All analyses were performed according to American Society for Bone and Mineral Research standards [53].

### 4.5. Cultivation of Primary Mouse Osteoblasts

Primary osteoblasts were isolated from long bones of 12-week–old mice, as previously described [54]. Briefly, long bones were harvested, and bone marrow was flushed out by centrifugation. Bone diaphyses were cut into pieces and subjected to a 60-min collagenase digestion using 125 U/mL collagenase type VI (Sigma-Aldrich, Taufkirchen, Germany). For osteoblast expansion, bone chips were cultivated in modified minimal essential medium (DMEM; Biochrom, Berlin, Germany) containing, 1% l-glutamine (PAN-Biotech, Aidenbach, Germany), 100 U/mL penicillin/streptomycin, and 0.5% amphotericin B (Fungizone), supplemented with 15% fetal calf serum (all from Gibco, Darmstadt, Germany) at 37 °C under 5% CO_2_. For the experiments, osteogenic differentiation was induced in the presence of 0.2 mmol/L ascorbate-2-phosphate and 10 mmol/L β-glycerophosphate (both from Sigma-Aldrich). Differentiation medium was replaced twice per week. Alkaline phosphatase stainings were performed by using a commercially available kit (Sigma-Aldrich) to confirm osteogenic differentiation after 14 days. In addition, mRNA samples were obtained after 10 days of osteogenic differentiation. Passage 2 osteoblasts were used for all experiments, which were performed in triplicates at least two times. The SensiFAST SYBR Hi-ROX One-Step Kit (Bioline, Memphis, TN, USA) was used according to the manufacturer’s guidelines to perform quantitative PCR. B2M (F: 5′-ATACGCCTGCAGAGTTAAGCA-3′, R: 5′-TCACAT GTCTCGATCCCAGT-3′) was used as the housekeeping gene. Relative gene expression of AP (F: 5′-GCT GAT CAT TCC CAC GTT TT-3′, R: 5′-GAG CCA GAC CAA AGA TGG AG-3′), OC (F: 5′-CTT GGT GCA CAC CTA GCA GA-3′, R: 5′-ACC TTA TTG CCC TCC TGC TT-3′, BSP (F: 5′-GAA GCA GGT GCA GAA GGA AC-3′, R: 5′-GAA ACC CGT TCA GAA GGA CA-3′), Runx2 (F: 5′-CCA CCA CTC ACT ACC ACA CG-3′, R: 5′-CAC TCT GGC TTT GGG AAG AG-3′), Col1 (F: 5′-GCT GCA TAC ACA ATG GCC TA-3′, R: 5′-TCA AGC ATA CCT CGG GTT TC-3′), and ERα (F: 5′-TCC GGC ACA TGA GTA ACA AA-3′, R: 5′-CCA GGA GCA GGT CAT AGA GG-3′) was calculated using the delta-delta CT method.

### 4.6. Statistics

Group size was *n* = 6–8 per group for bone phenotyping. Data were tested for normal distribution using the Shapiro–Wilk test; most data sets were normally distributed. Statistical testing was performed using t-tests with GraphPad Prism 8.4.3 (GraphPad Software, La Jolla, CA, USA). The level of significance was set at *p* ≤ 0.05. Results are presented as box-and-whisker plots (with median and interquartile range) from maximum to minimum, showing all data points.

## Figures and Tables

**Figure 1 ijms-23-02902-f001:**
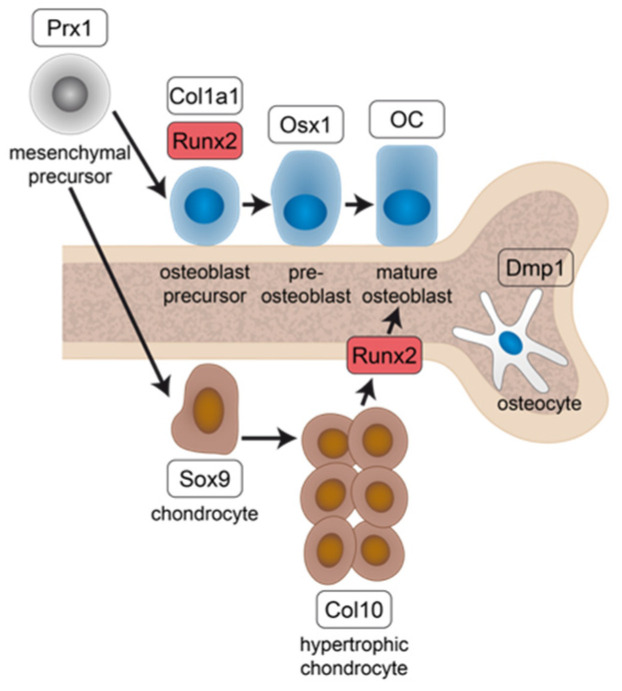
Schematic representation of osteoblast and chondrocyte lineage commitment from mesenchymal precursors, highlighting the progressive stages of differentiation with their associated characteristic genes (white boxes) that are used to develop osteoblast-specific Cre mouse lines (Table 1). Prx1 = paired-related homeobox 1; Runx2 = runt-related transcription factor 2; Osx1 = osterix; OC = osteocalcin; Dmp1 = dentin matrix protein 1; Sox9 = SRY-Box Transcription Factor 9; Col 10 = collagen type 10.

**Figure 2 ijms-23-02902-f002:**
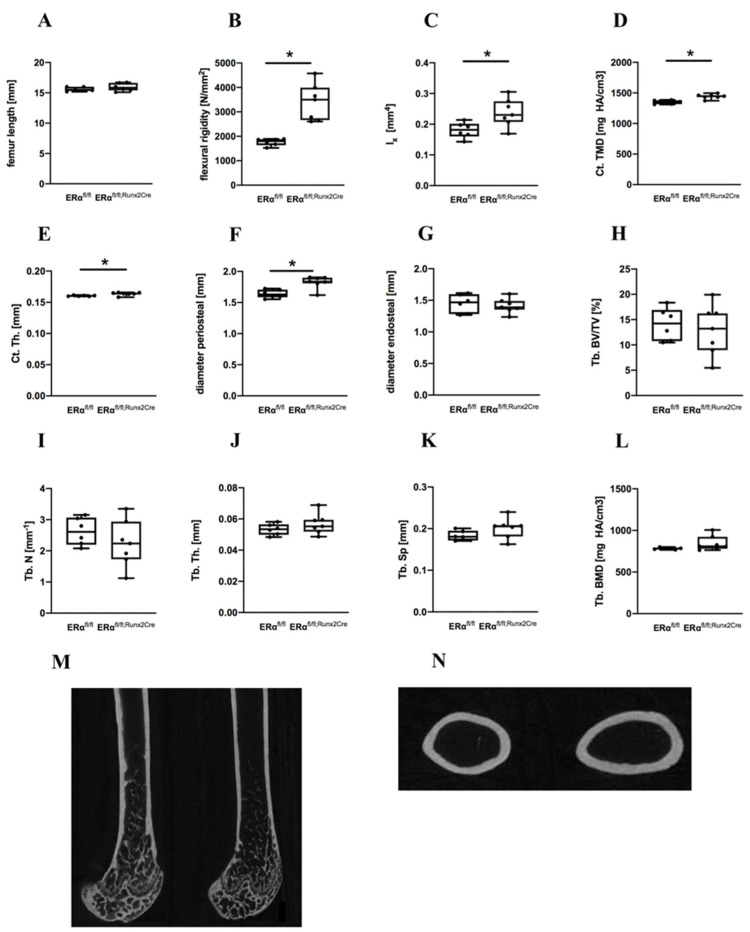
Bone-specific parameters of male ERα^fl/fl^ and ERα^fl/fl; Runx2Cre^ mice. (**A**) Femur length; (**B**) bending stiffness of the right femur; (**C**) moment of inertia; (**D**) cortical tissue mineral density (Ct. TMD); (**E**) cortical thickness (Ct. Th.); (**F**) diameter periosteal; and (**G**) diameter endosteal; (**H**) trabecular bone volume per tissue volume (Tb. BV/TV) (**I**) trabecular number (Tb. N). (**J**) Trabecular thickness (Tb. Th.); (**K**) trabecular separation (Tb. Sp). (**L**) Trabecular bone mineral density (Tb. BMD). (**M**) Representative μCT reconstructions of the right femurs representing the bones of ERα^fl/fl^ mice (left panel) and of ERα^fl/fl; Runx2Cre^ mice (right panel). (**N**) Representative cross-sectional μCT images of ERα^fl/fl^ mice (left panel) and ^ERαfl/fl; Runx2Cre^ mice (right panel). Data are shown as box-and-whisker plot (with median and interquartile range) from maximum to minimum, showing all data points. * Indicates significant effects with *p* ≤ 0.05.

**Figure 3 ijms-23-02902-f003:**
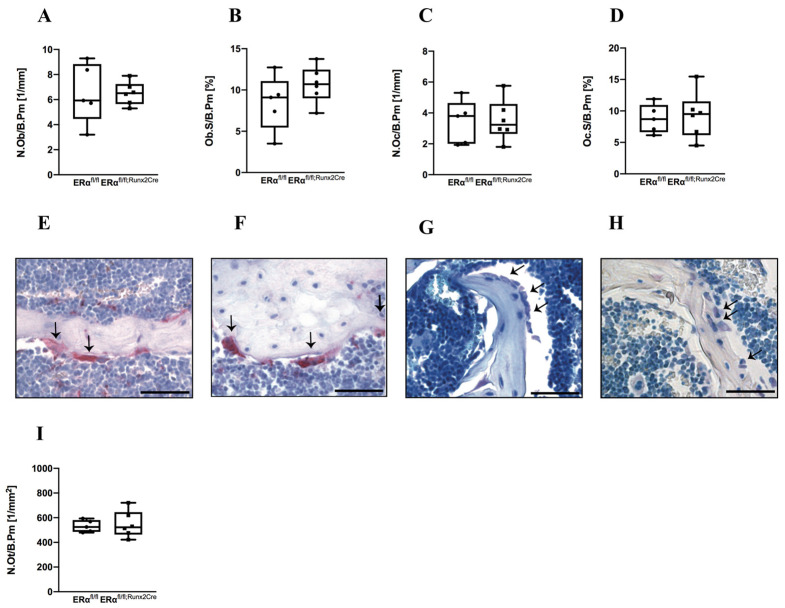
Histomorphometrical analysis of male ERα^fl/fl^ and ERα^fl/fl; Runx2Cre^ mice. Representative paraffin sections of femora were stained with TRAP or Toluidine blue and histomorphometrical analysis was performed to determine (**A**) number of osteoblasts per bone perimeter (N.Ob/B.Pm), (**B**) osteoblast surface per bone surface (Ob.S/B.Pm), (**C**) number of osteoclasts per bone perimeter (N.Oc/B.Pm), and (**D**) osteoclast surface per bone surface (Oc.S/B.Pm). Representative images of TRAP-stained sections from (**E**) ERα^fl/fl^ mice or (**F**) ERα^fl/fl; Runx2Cre^ mice as well as Toluidine blue-stained sections from (**G**) ERα^fl/fl^ mice or (**H**) ERα^fl/fl; Runx2Cre^ mice. (**I**) Number of osteocytes per bone perimeter (N.Ot/B.Pm). Scale bar = 50 μm. Data are shown as box-and-whisker plot (with median and interquartile range) from maximum to minimum, showing all data points.

**Figure 4 ijms-23-02902-f004:**
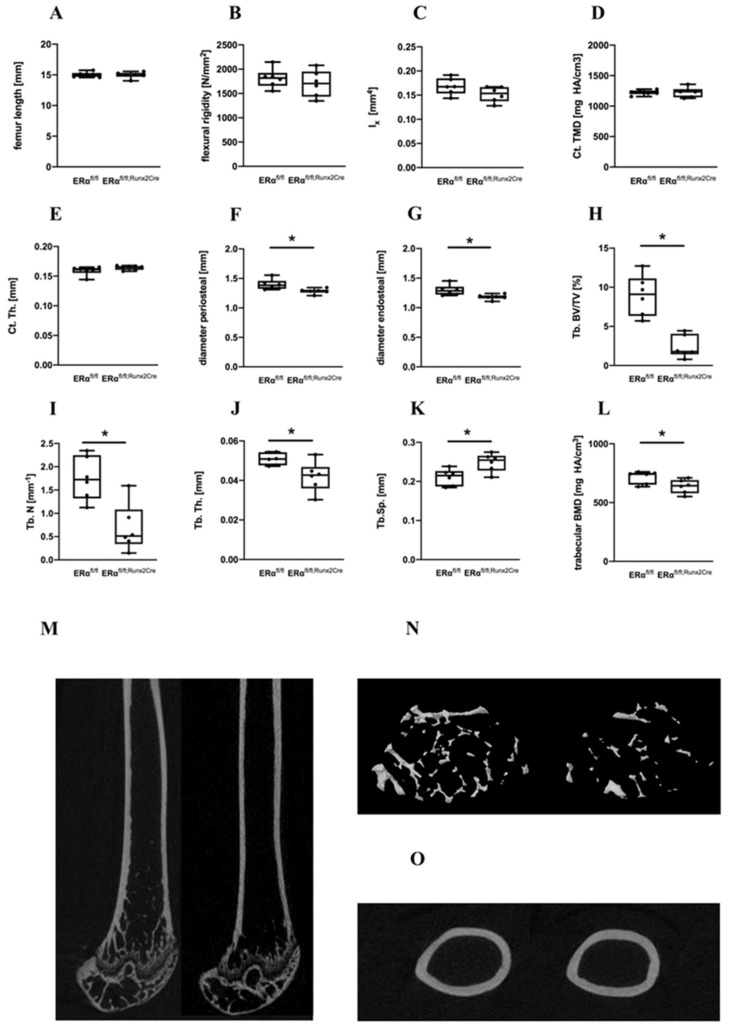
Bone-specific parameters of female ERα^fl/fl^ and ERα^fl/fl; Runx2Cre^ mice. (**A**) Femur length; (**B**) bending stiffness of the right femur; (**C**) moment of inertia; (**D**) cortical tissue mineral density (Ct. TMD); (**E**) cortical thickness (Ct. Th.); (**F**) diameter periosteal; and (**G**) diameter endosteal; (**H**) trabecular bone volume per tissue volume (Tb. BV/TV) (**I**) trabecular number (Tb. N). (**J**) Trabecular thickness (Tb. Th.); (**K**) trabecular separation (Tb. Sp); (**L**) trabecular bone mineral density (Tb. BMD). (**M**) Representative μCT reconstructions of the right femora representing the bones of ERα^fl/fl^ mice (left panel) and of ERα^fl/fl; Runx2Cre mice^ (right panel). (**N**) Representative μCT images of trabecular bone microarchitecture in the distal femurs of ERα^fl/fl^ mice (left panel) and ERα^fl/fl; Runx2Cre^ mice (right panel). (**O**) Representative cross-sectional μCT images of ERα^fl/fl^ mice (left panel) and ERα^fl/fl; Runx2Cre^ mice (right panel). Data are shown as box-and-whisker plot (with median and interquartile range) from maximum to minimum, showing all data points. * Indicates significant effects with *p* ≤ 0.05.

**Figure 5 ijms-23-02902-f005:**
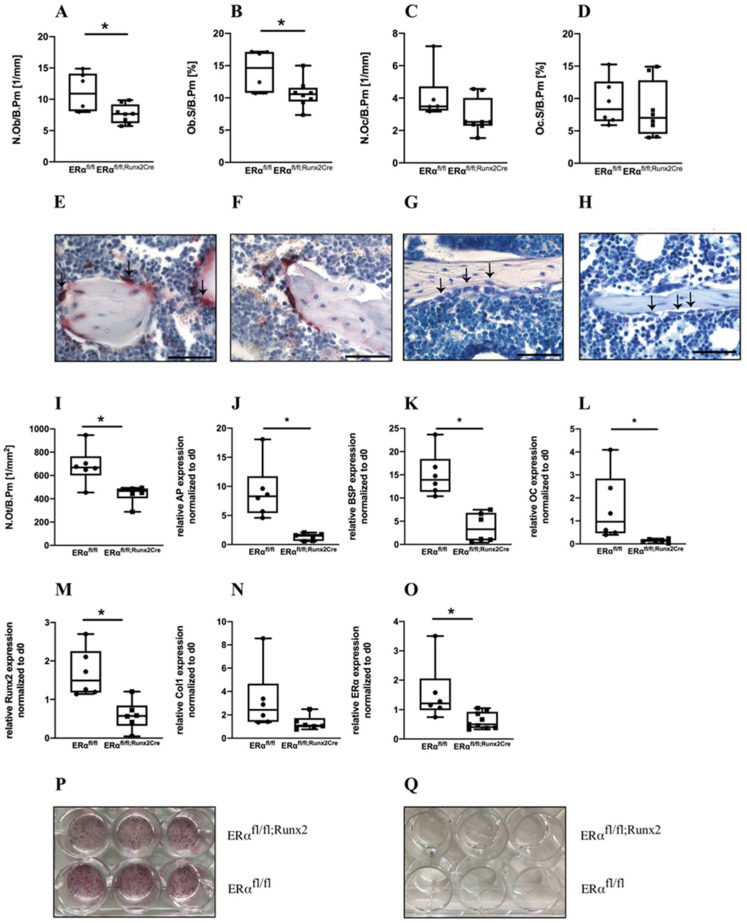
Histomorphometrical analysis of female ERα^fl/fl^ and ERα^fl/fl; Runx2Cre^ mice and differentiation ability of isolated primary osteoblasts. Representative paraffin sections of femurs were stained with TRAP or Toluidine blue and histomorphometrical analysis was performed to determine (**A**) number of osteoblasts per bone perimeter (N.Ob/B.Pm); (**B**) osteoblast surface per bone surface (ObS/B.Pm); (**C**) number of osteoclasts per bone perimeter (N.Oc/B.Pm); and (**D**) osteoclast surface per bone surface (OcS/B.Pm). Representative images of TRAP-stained sections from (**E**) ERα^fl/fl^ mice or (**F**) ERα^fl/fl; Runx2Cre^ mice as well as Toluidine blue-stained sections from (**G**) ERα^fl/fl^ mice or (**H**) ERα^fl/fl; Runx2Cre^ mice. (**I**) Number of osteocytes per bone perimeter (N.Ot/B.Pm). Primary osteoblasts were isolated from both ERα^fl/fl^ and ERα^fl/fl; Runx2Cre^ mice and relative AP expression was determined with (**J**) qPCR and by (**P**,**Q**) AP stainings in either differentiation medium at d10 (**P**) or control medium at d0 (**Q**). Relative expression of (**K**) BSP, (**L**) OC, (**M**) Runx2, (**N**) Col1, and (**O**) ERα was determined with qPCR. Scale bar = 50 μm. Data are shown as box-and-whisker plot (with median and interquartile range) from maximum to minimum, showing all data points. * Indicates significant effects with *p* ≤ 0.05.

**Table 1 ijms-23-02902-t001:** Effects of osteoblast-specific ERα knockout models on cortical and trabecular bone in male and female mice.

Mouse Model	Females		Males	
	Cortical	Trabecular	Cortical	Trabecular
Osteoblast lineage				
ERα ^fl/fl^ *Prx1*-Cre [29]	↓ Ct. Th.	↔	↓ Ct. Th.	↔
ERα ^fl/fl^ *Runx2*-Cre [30]	N.a.	↓	N.a.	N.a.
ERα ^fl/fl^ *Runx2*-Cre	↔	↓	↑	↔
ERα ^fl/fl^ *Col1a1*-Cre [29]	↔	↔	↔	↔
ERα ^fl/fl^ *Osx1*-Cre [29]	↓	↔	N.a.	N.a.
ERα ^fl/fl^ *OC*-Cre [31]	↓	↓	↔	↔
ERα ^fl/fl^ *OC*-Cre [32]	↓	↓	N.a.	N.a.
ERα ^fl/fl^ *OC*-Cre [33] C57Bl/6 background	↓	↓	↑ Ct. Ar. ↑ Femoral length	↑ Tb. Th, BV/TV
ERα ^fl/fl^ *OC*-Cre [34]	↓	↓ Tb. BV	↔	↔
ERα ^fl/fl^ *Dmp1*-Cre [35]	↔	↔	↔	↓

↓ = reduced; ↑ = increased; ↔ = unaltered; N.a.= not available; Ct. Th. = cortical thickness, Ct. Ar. = cortical area; Tb. Th = trabecular thickness, BV/TV = relative bone volume; Tb. BV = trabecular bone volume.

## Data Availability

Not applicable.

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
