# Peer review of "Bone Mass and Osteoblast Activity Are Sex-Dependent in Mice Lacking the Estrogen Receptor α in Chondrocytes and Osteoblast Progenitor Cells"

_ijms, 2022, doi:10.3390/ijms23052902_

Round 1

Reviewer 1 Report

In this manuscript, the authors investigated the role of ERα in both osteoblast lineage-specific cells and hypertrophic chondrocytes by using the Runx2-Cre mouse model. It brings us a little but important step on the regulatory functions of ERα in bone development. I would like to put forward the following questions and comments:

  1. In the statistics, it is mentioned that “Group size was n=6-8 per group for bone phenotyping”, how many samples/mice were included in each group for the biomechanical testing and CT scanning? Did bilateral femur were included or just one side?
  2. Could you explain why femur rather than other bones were studied?

Reviewer 2 Report

In the present study authors have conducted a throrough analysis of bone structural parameters and properties in mice lacking ERα in hypertrophic chondrocytes and osteoblast progenitors. Depending on the promoter used, CRE-based ERα knockouts have been displaying various cortical and trabecular bone phenotypes. Utilizing the Runx2 promoter here, authors add to the present literature by evaluating effects in both osteoblast precursors and transdifferentiating chondrocytes.

My comments and suggestions, in no order of importance, are the following:

1) Title: It would be useful to include the main result of the study into the title, e.g. "Bone mass and osteoblast activity are sex-dependent in mice lacking ERα in hypertrophic chondrocytes and osteoblast progenitors"

2) Authors suggest that bone development from endochondral ossification (i.e. through chondrocyte hypertrophy) might be particularly altered in Runx2-driven deletion. Wouldn't it therefore be interesting to analyze both primary (as performed) and secondary ossification centers (epiphyseal femur)?

3) Did authors perform specific analyses at the growth plate? It might be interesting to analyze growth plate thickness and osteoclast activity in this subregion.

4) The data representation (box & whiskers with IQR and median) is not ideal for these small data sets. As the majority of data sets were normally distributed, I'd suggest using bar graphs indicating the mean (with or without SD/SEM) in combination with individual data points.

5) Please check whether the significance in Figure 2E is true. A difference in means can hardly be discerned here, while the SD is indeed very small.

6) Please check and use the ASBMR nomenclature for histomorphometric analyses in 2D. Osteoblast/clast number = N.Ob/N.Oc. Surface cannot be calculated in 2D, but should be bone perimeter. Numbers normalized by perimeter are: Oc.Pm and Ob.Pm. Alternatively these can be normalized for bone area. Oc.BAr and Ob.BAr.

7) Did authors perform ALP staining (Fig 5M) in osteoblasts maintained in control medium? Alizarin red staining for mineralized nodules are typically performed to assess differentiation capacity.

8) Bsp and Oc are late markers of osteoblast differentiation. At day 10 it would be more interesting to look at Runx2 and Col1a1, this should be performed when authors have validated primers.

9) Have authors verified ERalpha expression in primary osteoblasts? It seems it would be important to verify that Runx2-ERalpha KOs indeed lack its expression in vitro.

10) Please include the following study in the discussion: Pubmed ID: 35123146.

The reviewer hopes comments and suggestions are deemed useful. 

Reviewer 3 Report

  1. The Introduction clearly needs to expand on the relevance of studying ERa expression and role in bone formation, and explain why one should investigate sex-specific differences in bone formation. The Introduction details on the findings, without providing a clear summary of the results, and the overall finding.
  2. The authors need to expand on the ERa/ERb differences in expression and role in bone function. Could those differences be sex-specific as well?
  3. The Discussion lacks cohesiveness. Edit the discussion to make the overall message clearer. Each paragraph should contain comparison of both male and female mice, according to individual features analyzed, instead of grouping observations into separate paragraphs based on male/female.
  4. Is there any physiological relevance for the observations? Do the ERa mediated differences in bone formation inform human bone development biology? The authors should comment on these points in the Discussion.

Round 2

Reviewer 2 Report

The reviewer thanks the authors for their efforts in revising the manuscript and providing an extensive rebuttal.

My queries have been adequately addressed.